# Distributed Learning Based Joint Communication and Computation Strategy of IoT Devices in Smart Cities

**DOI:** 10.3390/s20040973

**Published:** 2020-02-12

**Authors:** Tianyi Liu, Ruyu Luo, Fangmin Xu, Chaoqiong Fan, Chenglin Zhao

**Affiliations:** 1International School, Beijing University of Posts and Telecommunications, Beijing 100876, China; 2School of Information and Telecommunication Engineering, Beijing University of Posts and Telecommunications, Beijing 100876, China; 3Key Laboratory of Universal Wireless Communications, Ministry of Education, Beijing University of Posts and Telecommunications, Beijing 100876, China

**Keywords:** smart city, Internet of Things, mobile edge computing, potential game, Q-learning

## Abstract

With the development of global urbanization, the Internet of Things (IoT) and smart cities are becoming hot research topics. As an emerging model, edge computing can play an important role in smart cities because of its low latency and good performance. IoT devices can reduce time consumption with the help of a mobile edge computing (MEC) server. However, if too many IoT devices simultaneously choose to offload the computation tasks to the MEC server via the limited wireless channel, it may lead to the channel congestion, thus increasing time overhead. Facing a large number of IoT devices in smart cities, the centralized resource allocation algorithm needs a lot of signaling exchange, resulting in low efficiency. To solve the problem, this paper studies the joint policy of communication and computing of IoT devices in edge computing through game theory, and proposes distributed Q-learning algorithms with two learning policies. Simulation results show that the algorithm can converge quickly with a balanced solution.

## 1. Introduction

With the increasing number of cities and urban population, there has been an increasing interest in the smart city. The concept of smart city emphasizes a solution that takes into account the sustainability of the city in all regions and project implementation phases [1]. With making full use of a large number of interconnected devices, smart cities have been attached to the Internet of Things (IoT) technology. In the smart city, the automation strategy based on massive IoT devices deployment to gather Big Data to get insights into city behavior to improve its services [2]. In addition, the term Internet of Things represents the ability of intelligence devices to sense, collect, share data across the Internet or a local network, and then act according to the received information [3]. With the rapid increase of smart connected objects (such as personal devices, sensors, actuators) in the smart city, different simultaneous interpreting of delay is required by different time critical IoT applications. For example, early warning systems or live event broadcasting, present particular challenges [4,5,6]. In addition, many IoT devices require enhanced computing performance, such as real-time camera identification. To sum up, how to improve the quality of service (QoS) to ensure the normal operation of smart city has become an important issue.

In order to meet this challenge, mobile edge computing (MEC) [7] can efficiently improve the QoS for applications that require intensive computations and low latency, which have been raised to deploy computing resources closer to end devices in the smart city [8]. Once the IoT devices have started being put online, a new step in the evolution of mobile networks was taken through the addition of edge and fog computing, where small nodes at the edge of the network take up some of the load on the cloud backend [6]. In addition, when new IoT devices are connected due to urban construction, MEC is very convenient and flexible because it adopts wireless access. As shown in Figure 1, various IoT devices in the smart city can connect to the MEC servers. Through the MEC server, the IoT devices can perform the computation offloading, thus reducing the computation time consumption and resulting in lower latency.

It is quite good to use MEC in the smart city, but resource allocation in the smart city is still a problem to be solved. First of all, if a large number of devices simultaneously do the computation offloading to the MEC server, due to the limited channel resources, the wireless network may become congested, thus increasing the time consumption. Second, IoT devices in smart cities are different from traditional sensors, many of which have some computing capability (such as Raspberry Pi [9]). If IoT devices adopt edge computing completely, this part of the computing resources will be wasted. Third, the smart city is generally composed of many parts, including many different services which has different requirements. A reasonable communications and computing strategy should ensure that as many IoT devices as possible are functioning properly, rather than having a small number of devices occupy all resources so that some parts of a smart city cannot work due to lack of resources. Moreover, the centralized resource allocation algorithm with a large amount of information collection and signaling exchange may not be suitable for smart city scenarios.

For the above problems and challenges, this paper focuses on how to perfectly do the resources (including wireless spectrum resources and computing resources) allocated in the smart city, mainly aiming at making as many IoT devices associated with one MEC server as possible accomplish the tasks in a low latency. We use distributed Q-learning to allocate resources and combine game theory to ensure the balance of solutions. In addition, we also creatively used two different learning policies. The main contributions of our propose scheme are as follows:Based on the system model of smart city and MEC, we formulate a computation offloading game with IoT devices in the smart city, and prove that the game has at least one Nash equilibrium point.Combined with game theory and system model, we proposed two distributed Q-learning algorithms with different learning policies to get the joint communication and computation strategies for each IoT devices in the smart city. The distributed Q-learning requires only a small amount of signaling exchange and has a good convergence performance, which is very suitable for smart cities. The combination of Q-learning and MEC can effectively improve the QoS of the smart city system.

The rest of this paper is organized as follows. In Section 2, we review the related work. In Section 3, we introduce our system model and describe our problem. In Section 4, we design a noncooperative game model for computation offloading, which can prove the existence of Nash equilibrium. In Section 5, we describe two stateless distributed Q-learning algorithms with different learning policies for edge computation offloading game. In Section 6, simulation results are provided, showing that the distributed Q-learning algorithm has good performance. The conclusions and future work are given in Section 7.

## 2. Related Work

In recent years, in order to make full use of the computing and spectrum resources, many studies of communication and computation strategies have proposed. Some methods are based on cloud or MEC, and a few are specific to the smart city.

There are different methods meeting the challenge of communication and computation in different application scenarios for different purposes. Some of those strategies are developed based on cloud computing. Štefanič has introduced a general SWITCH architecture with its subsystems, TOSCA orchestration standard, and software engineering workflow in SWITCH to address entire life cycle of time-critical cloud applications [4]. Zeng has presented an architecture of multiple cloud service providers (CSPs) or “Cloud-of-Clouds” to provide services to the continuous writing applications (CWA) by using a novel resource scheduling algorithm to minimize the cost of entire systems [10]. Furthermore, it is very popular to use mobile edge computing (MEC) to slove communication and computation problems. Some have proposed the centralized algorithm, which needs to upload all information to a central node. Huang has proposed a control scheme for offloading vehicular communication traffic in the cellular network to vehicle-to-vehicle (V2V) paths using software-defined network (SDN) inside the MEC architecture [11]. Berno has considered a model for the allocation of processing tasks in MEC, and its allocation problem is formulated as a centralized (offline) optimization program with delay constraints (deadlines) [12]. Meanwhile, the distributed algorithm is also worthy of attention. T. Q. Dinh proposed a learning method for computation offloading in MEC, which is based on non-cooperative game and it focuses on maximizing CPU cycles [13]. Chen designed a distributed computation offloading algorithm that can achieve a Nash equilibrium for mobile-edge cloud computing [14]. Ranadheera summarized the previous work and compared the characteristics and advantages of different distributed algorithms in MEC [15]. Most of these distributed algorithms are based on game theory, giving us a lot of inspiration.

Considering the guarantees for low-latency applications in smart cities and the limited spectrum-computation and sub-channel resource, traditional cloud computing faces great challenges [16,17,18]. Thus, people have proposed several ways to complete the shortcomings of cloud computing in the smart city. Some solutions are still based on cloud computing. Kakderi has focused on the storm cloud paradigm as a solution to deploy a portfolio of smart cities applications on a single cloud-based platform and migrating existing applications to the cloud environment using the platform and its accompanied tools [5]. Ciobanu aimed at solving the problem of offloading data and computing from mobile devices to cloud, to fog nodes, or to other nearby mobile devices. The novelty of the proposal is to add a layer composed exclusively of mobile devices that collaborate in an opportunistic fashion [6]. There are also schemes based on data fusion. Facing the large amount of data generated by IoT devices, Esposito proposed an event-based data fusion approach, which effectively utilizes limited resources [19]. Moreover, there are also some studies based on MEC. Sapienza proposed a solution to detect abnormal or critical events such as terrorist threats, natural, and man-made disasters by using MECs. The proposed solution allows cooperation among the Base Transceiver Stations to rapidly notify the users which are close to the critical area [20]. Zhao proposed that edge servers could be set up in smart cities to improve QoS, and provided two methods of edge resource allocation, Enumeration-Based algorithm, and Clustering-Based algorithm [21]. Deng proposed four methods to schedule tasks for computation-intensive and time-sensitive smart city applications with the assistance of IoT based on multi-server mobile edge computing [22]. However, for the smart city scenario, studies based on learning and game theory are rare.

To sum up, there have been many previous attempts to solve resource allocation problems in smart cities, but few are based on game theory and learning algorithms. Therefore, we improved the Q-learning algorithm based on MEC, and proposed two methods with different learning policies for smart city.

## 3. System Model

Smart city is a huge system. In the smart city, a large number of services running over massive end IoT devices as well as applications hosted in remote servers. These IoT devices come in a wide variety of categories and services. It is very difficult to study the model of the whole smart city. Thus, we only consider one block in the smart city. In this paper, the uplink of wireless network integrated with multiple IoT devices is mainly considered. Furthermore, we aim at making as many IoT devices associated with one MEC server as possible meet the threshold and minimizing the total time delay. Thus, in the following article, in a block of the smart city, it is assumed that IoT devices in this block are associated with one MEC server. The set of IoT devices is presented by $M=\left\{1,2,\cdots ,M\right\}$. Moreover, the set of IoT devices associated with the MEC server shares $N=\left\{1,2,\cdots ,N\right\}$ orthogonal sub-channels. In the following article, we will show that the communication and computation model for the IoT devices in the smart city, and then describe our main problem.

### 3.1. Communication Model

First, we introduce the communication model for wireless access of the IoT devices. It is assumed that the distance between the IoT device *m* and its MEC server is $d_{m}$. For the links between IoT devices and MEC server, the sub-channel gain from the MEC server to IoT device *m* is defined as $h_{m}$. In addition, in our model, we assume that the sub-channel gains $h_{m}$ are composed of sub-channel path loss PL and shadow fading. We assume that, for all IoT devices, the log-normal shadowing fading $g_{m}$ is independent with zero mean and 8 dB standard derivation. $PL(d_{m})$ represents the path loss at a reference distance $d_{m}$. Thus, the sub-channel gain can be computed as:(1)$$h_{m}=PL\left(d_{m}\right)g_{m}$$

In addition to meet the goal, optimal sub-channel allocation is important to support the IoT devices so that we need to consider the interference between the IoT devices. It is assumed that the transmit power of the IoT device *m* associated with the MEC server is denoted by $P_{m}$. $I_{m}^{n}$ is the interference between the IoT device *m* and other IoT devices which share the same sub-channel *n*. It can be expressed as:(2)$$I_{m}^{n}=\sum_{i\in M\backslash \left\{m\right\}}\zeta_{i}^{n}P_{i}h_{i}$$

In particular, $I_{m}^{0}$ is equal to zero for all devices. In addition, $\zeta_{i}^{m}$ is the sub-channel assignment indicator, and:(3)$$\zeta_{m}^{n}=\left\{0,1\right\}\;\forall m,n$$
(4)$$\sum_{n=1}^{N}\zeta_{m}^{n}=1\;\forall m$$
where $\zeta_{m}^{n}=1$ indicates that the device *m* use the sub-channel *n*. According to Shannon formula, the instantaneous rate of the IoT device *m* on sub-channel *n* can be formulated as
(5)$$r_{m}^{n}=W_{n}\log_{2}\left(1+\frac{P_{m}h_{m}}{\delta^{2}+I_{m}^{n}}\right)$$In the above formulate, $\delta^{2}$ is the thermal noise, $W_{n}$ is the sub-channel bandwidth.

### 3.2. Computation Model

Then, we introduce the computation model. In the smart city system, it is assumed that each IoT device will handle a specific type of task in a period time. The set of computation tasks is J. For the IoT device *m*, it has a kind of computation task $J_{m}\in J$ and $J_{m}=\left(D_{m},C_{m},TH_{m}\right)$, where $D_{m}$ is the amount of input data for the task, including all parameters and codes. $C_{m}$ is the number of CPU cycles required to accomplish the computation task. In addition, $TH_{m}$ is a time threshold for task $J_{m}$. If $T_{m}\leq TH_{m}$, task $J_{m}$ is considered to be successfully completed. For convenience of representation, we refer to the IoT device *m* in this case as the device that meets the threshold. Otherwise, task $J_{m}$ is overtime. The device *m* will accomplish the task locally or in the edge. Both approaches are discussed as follows.

#### 3.2.1. Local Computing

For the local computing approach, IoT device *m* will execute task $J_{m}$ locally by its own computation ability. Different devices have different computation capabilities in a smart city system. For local computing, let $f_{m}^{l}$ be the computation capability (i.e., CPU cycles per second) of the IoT device *m*. It should be noted that, when designing a smart city system, the designer should consider the computation capability of the IoT devices. The devices need to have appropriate computing capability to meet their services and ensure that the system is not congested. In other words, it can be considered that the probability of queueing when the task is locally computed is very low. Therefore, in order to simplify the model, the queueing delay of tasks in local computation can be ignored. The computation execution time of the task $J_{m}$ by local computing is then given as:(6)$$T_{m}^{l}=\frac{C_{m}}{f_{m}}$$

#### 3.2.2. Edge Computing

For the edge computing, IoT device *m* will offload its task $J_{m}$ to the nearby MEC server. Like some studies such as [23], for simplifying the problem, only one MEC server is deployed in one block of the city. In addition, our study focuses on one block. Thus, it is assumed that there is only one MEC server in the model with computation capability $f^{e}$. Therefore, the time for the MEC server to execute a task is:(7)$$t_{m}^{e}=\frac{C_{m}}{f^{e}}$$

When offloading a task, the device first selects a wireless sub-channel and uploads all data using 4G/5G or other wireless approaches to the MEC server. The time required for IoT device *m* uploading a task $J_{m}$ with sub-channel *n* is
(8)$$t_{m,n}^{u}=\frac{D_{m}}{r_{n}^{m}}$$

When the upload is complete, the task will be processed immediately. Based on the above formula, we can obtain the total expected time for IoT device *m* to complete a task $J_{m}$ with sub-channel *n* using edge computing, which is expressed as follows:(9)$$T_{m}^{e}=t_{m,n}^{u}+t_{m}^{e}$$

Based on the above system model, we can estimate the time overhead for the IoT devices to accomplish tasks with edge computing and local computing. Similar to some studies such as [14], we only consider IoT devices uploading input data to the MEC server. The time required to transmit the output data is neglected in this model because the size of the output data in general is much smaller than the size of the input data, and the download rate is much faster compared to the upload rate. In addition, in the study of smart cities, wireless channel resources and computing resources are usually the scarcest and should be paid most attention to.

### 3.3. Problem Description

In the smart city system, MEC server and IoT devices generally have stable power supply, so we can neglect the power consumption of IoT devices and MEC server. It is more important to consider the time overhead of the task. By offloading tasks to the MEC server, IoT devices can effectively reduce time consumption. However, if too many IoT devices simultaneously choose to offload the computation tasks to the MEC server via the same wireless sub-channel, it may lead to severe sub-channel interference, resulting in long upload time. Therefore, finding the proper strategies adopted by each device is the key problem.

In the system, the processing latency of accomplishing tasks need to be minimized. For IoT device *m*, the expected time to finish task $J_{m}$ is:(10)$$T_{m}=\left(1-s_{m}\right)*T_{m}^{l}+s_{m}*T_{m}^{e}$$
where $s_{m}$ is the computation state of device *m*. If $s_{m}=1$, the device chooses edge computing. Otherwise, the device finishes its task locally. We want to accomplish tasks successfully as many as possible and spend as little time overhead as possible on each task. Therefore, for one device *m*, our study target is to find a proper computing strategy (including computing state and wireless sub-channel selection) to minimize $T_{m}$ for meeting the threshold $TH_{m}$. We consider that every IoT device is selfish and want to reduce their task processing latency. Based on this condition, a game-theory based model will be introduced to formulate and solve the problem.

## 4. Non-Cooperative Game Model for Computation Offloading

According to the problem description in Section 3, it is shown that the computation decisions among IoT devices are coupled. The time overhead of a device may be influenced by the decision of other devices. However, in the distributed decision-making system, each user’s decision is independent. In this case, similar to some studies [13,24], non-cooperative game theory can be applied to the problem. In the non-cooperative game, each IoT device is selfish and they want to minimize their time overhead by making decision independently. In this section, we will show the game formulation and use the potential game to analyze the Nash equilibrium.

### 4.1. Game Formulation

Based on the system model in Section 3, for the IoT device *m*, we define its action as follows:(11)$$a_{m}=\left\{\begin{array}{cc} \sum_{i=1}^{N}\left(i\zeta_{m}^{i}\right) & s_{m}=1\\ 0 & s_{m}=0 \end{array}\right.$$

The action $a_{m}\in A=\left\{0,1,\ldots ,N\right\}$ represents the computational offloading decision for device *m*. It means that, when the device *m* choose edge computing, $a_{m}$ is equal to the sub-channel index; otherwise, $a_{m}$ is zero. Note that all the IoT devices have a same action space, so they can share the action set A. For all the devices, the actions form an vector $a=\left\{a_{1},a_{2},\ldots a_{m}\right\}$ representing the action vector (or strategy vector) of the devices. Let a special action vector $a_{-m}=\left\{a_{1},\ldots ,a_{m-1},a_{m+1},\ldots ,a_{M}\right\}$ be the actions decided by all other IoT devices except device *m*. Combining the other devices’ actions $a_{-m}$, the device *m* can select a proper action $a_{m}$ to minimize the time overhead. Then, we can formulate the problem as:(12)$$F:\min_{a_{m}\in A}U_{m}\left(a_{m},a_{-m}\right)$$
where $U_{m}\left(a_{m},a_{-m}\right)$ is called utility function. It can be expressed as:(13)$$U_{m}\left(a_{m},a_{-m}\right)=T_{m}=\left\{\begin{array}{cc} T_{m}^{e} & a_{m}>0\\ T_{m}^{l} & a_{m}=0 \end{array}\right.$$

Based on the above formula, we can formulate the problem as a strategic form game as the following definitions:

**Definition** **1.**
*Edge Computation Offloading Game (ECOG): We define a strategic form game Γ=(M,A,U) as an edge computation offloading game, where*

*M is the set of IoT devices in the smart city, which are the players in the game.*

*A is the action set shared by all the IoT devices.*

*$U={\left\{U_{m}\right\}}_{m\in M}$ is the set of utility functions for all devices. $U_{m}$ is defined in (Equation 13).*



More definitions and proofs on game theory and strategic form game are given by [25]. Now, the definition of Nash equilibrium is as follows, which accounts for a steady state of an ECOG:

**Definition** **2.**
*Nash Equilibrium (NE): A strategy vector $a=\left\{a_{1}^{*},a_{2}^{*},\ldots a_{m}^{*}\right\}$ is a Nash equilibrium of the ECOG if there is no IoT device can further reduce its utility by changing action unilaterally, i.e.:*
(14)$$U_{m}\left(a_{m}^{*},a_{-m}^{*}\right)\leq U_{m}\left(a_{m},a_{-m}^{*}\right)\;\forall a_{m}\in A,\forall m\in N$$


Nash equilibrium is an important concept in game theory. For our problem, it has great meaning to the solution. If a device *m* chooses edge computing when the game is at NE, it must reduce the time overhead by computation offloading. Because if device *m* chooses edge computing and it spends more time to finish the task, it can change to local computing to decrease its utility, which is against the definition of a Nash equilibrium. In other words, the NE point must be a relatively good solution to the problem described in Section 3. Due to the concept of the Nash equilibrium, the IoT devices in NE can achieve a mutually satisfactory solution and no one has the intention to deviate the steady state.

Although NE is very important, we still don’t know whether NE exists in ECOG. In the next part, a powerful tool, potential game will be introduced to prove the existence of Nash equilibrium.

### 4.2. Potential Game

Potential game is a strategic form game with a finite number of players. The definition of the ordinary potential game is as follows:

**Definition** **3.**
*Ordinary Potential Game (OPG): A strategic form game is called ordinary potential game (OPG) if it admits a potential function $\Phi_{m}\left(a\right)$ such that, for every m∈M, $a_{-m}=\left\{a_{1},\ldots ,a_{m-1},a_{m+1},\ldots ,a_{M}\right\}$ and $a_{m},a_{m}^{\prime }\in A$, if*
(15)$$U_{m}\left(a_{m}^{\prime },a_{-m}\right)<U_{m}\left(a_{m},a_{-m}\right)$$
*we have*
(16)$$\Phi_{m}\left(a_{m}^{\prime },a_{-m}\right)<\Phi_{m}\left(a_{m},a_{-m}\right)$$


OPG has an appealing property that every ordinary potential game always has a pure strategy Nash equilibrium and the finite improvement property (FIP) [26]. We can use OPG to prove that the Nash equilibrium of ECOG exists, that is, proving ECOG is OPG. Before processing, we show the following result at first:

**Lemma** **1.**
*When the IoT device m chooses to select edge computing using wireless sub-channel n, its interference $I_{m}^{n}$ must satisfied that $I_{m}^{n}<H_{m}$, with the threshold*
(17)$$H_{m}=\frac{P_{m}h_{m}}{2^{\frac{D_{m}}{W_{m}\left(T_{m}^{l}-t_{m}^{e}\right)}}-1}$$


**Proof.** Since device *m* chooses sub-channel *n* to do the tasks, the time overhead of edge computing must less than local computing, so we know that $T_{m}^{e}<T_{m}^{l}$. According to Section 3, this can be expressed as:
(18)$$I_{m}^{n}<\frac{P_{m}h_{m}}{2^{\frac{D_{m}}{W_{m}\left(T_{m}^{l}-t_{m}^{e}\right)}}-1}=H_{m}$$Note that there is a very small probability that $T_{m}^{e}$ is going to be equal to $T_{m}^{l}$. In that case, we assume that the device will choose local computing because it is more convenient for no uploading data. Then, Lemma 1 has been proved. □

Based on Lemma 1, we can determine the potential function of ECOG:

**Theorem** **1.**
*The edge computation offloading game is an ordinary potential game, which has at least one NE and the finite improvement property, with the potential function:*
(19)$$\Phi_{m}\left(a\right)=\sum_{i=1}^{M}P_{i}h_{i}I_{i}^{a_{i}}+2\sum_{i=1}^{M}\left(1-s_{i}\right)P_{i}h_{i}H_{i}$$


**Proof.** Firstly, supposing that a IoT device *m* changes its action from $a_{m}$ to $a_{m}^{\prime }$ to reduce its utility, which is $U_{m}\left(a_{m}^{\prime },a_{-m}\right)<U_{m}\left(a_{m},a_{-m}\right)$. There are three cases to achieve this reduction: (1) $a_{m},a_{m}^{\prime }>0$, (2) $a_{m}=0,a_{m}^{\prime }>0$, and (3) $a_{m}>0,a_{m}^{\prime }=0$.For case (1), $a_{m},a_{m}^{\prime }>0$ means that the device *m* chooses edge computing but changes the sub-channel to do the computation offloading. According to our hypothesis, the utility is decreased, such that $U_{m}\left(a_{m}^{\prime },a_{-m}\right)<U_{m}\left(a_{m},a_{-m}\right)$. Since device *m* just does the sub-channel changing, the processing time is no change. It implies that only an increase in the upload rate results in a reduction in time overhead:
(20)$$W_{m}\log_{2}\left(1+\frac{P_{m}h_{m}}{\delta^{2}+I_{m}^{a_{m}}}\right)<W_{m}\log_{2}\left(1+\frac{P_{m}h_{m}}{\delta^{2}+I_{m}^{a_{m}^{\prime }}}\right)$$It is easy to obtain that:
(21)$$I_{m}^{a_{m}^{\prime }}<I_{m}^{a_{m}}$$It means that the interference for device *m* is reduced, and devices that are computing locally are not affected. According to (Equation 19) and the definition of potential function, we know that:
(22)$$\begin{array}{cc} \; & \Phi_{m}\left(a_{m}^{\prime },a_{-m}\right)-\Phi_{m}\left(a_{m},a_{-m}\right)=\sum_{i=1,a=\left\{a_{m}^{\prime },a_{-m}\right\}}^{M}P_{i}h_{i}\sum_{j=1,j\neq i}^{M}\zeta_{j}^{a_{i}}P_{j}h_{j}-\sum_{i=1,a=\left\{a_{m},a_{-m}\right\}}^{M}P_{i}h_{i}\sum_{j=1,j\neq i}^{M}\zeta_{j}^{a_{i}}P_{j}h_{j}\\ \; & =P_{m}h_{m}\sum_{i=1,i\neq m}^{M}\zeta_{i}^{a_{m}^{\prime }}P_{i}h_{i}+\sum_{i=1,i\neq m}^{M}P_{i}h_{i}\left(\sum_{j=1,j\neq i,j\neq m}^{M}\zeta_{j}^{a_{i}}P_{j}h_{j}+\zeta_{i}^{a_{m}^{\prime }}P_{m}h_{m}\right)\\ \; & \;-P_{m}h_{m}\sum_{i=1,i\neq m}^{M}\zeta_{i}^{a_{m}}P_{i}h_{i}-\sum_{i=1,i\neq m}^{M}P_{i}h_{i}\left(\sum_{j=1,j\neq i,j\neq m}^{M}\zeta_{j}^{a_{i}}P_{j}h_{j}+\zeta_{i}^{a_{m}}P_{m}h_{m}\right)\\ \; & =P_{m}h_{m}\sum_{i=1,i\neq m}^{M}\zeta_{i}^{a_{m}^{\prime }}P_{i}h_{i}-P_{m}h_{m}\sum_{i=1,i\neq m}^{M}\zeta_{i}^{a_{m}}P_{i}h_{i}+P_{m}h_{m}\sum_{i=1,i\neq m}^{M}\zeta_{i}^{a_{m}^{\prime }}P_{i}h_{i}-P_{m}h_{m}\sum_{i=1,i\neq m}^{M}\zeta_{i}^{a_{m}}P_{i}h_{i}\\ \; & =2P_{m}h_{m}\left(I_{m}^{a_{m}^{\prime }}-I_{m}^{a_{m}}\right) \end{array}$$Because of (Equation 21), the above expression indicates that $\Phi_{m}\left(a_{m}^{\prime },a_{-m}\right)<\Phi_{m}\left(a_{m},a_{-m}\right)$. Then, we use the same way to prove case (2). For case (2), we can firstly subtract the two potential functions:
(23)$$\begin{array}{cc} \; & \Phi_{m}\left(a_{m}^{\prime },a_{-m}\right)-\Phi_{m}\left(a_{m},a_{-m}\right)=\sum_{i=1,a=\left\{a_{m}^{\prime },a_{-m}\right\}}^{M}P_{i}h_{i}I_{i}^{a_{i}}-\sum_{i=1,a=\left\{a_{m},a_{-m}\right\}}^{M}P_{i}h_{i}I_{i}^{a_{i}}-2P_{m}h_{m}H_{m}\\ \; & =P_{m}h_{m}I_{m}^{a_{m}^{\prime }}-2P_{m}h_{m}H_{m}+\sum_{i=1,i\neq m,a=\left\{a_{m}^{\prime },a_{-m}\right\}}^{M}P_{i}h_{i}I_{i}^{a_{i}}-\sum_{i=1,i\neq m,a=\left\{a_{m},a_{-m}\right\}}^{M}P_{i}h_{i}I_{i}^{a_{i}}\\ \; & =2P_{m}h_{m}\left(I_{m}^{a_{m}^{\prime }}-H_{m}\right) \end{array}$$
where $H_{i}^{\prime }$ is the threshold of $a=\left\{a_{m},a_{-m}\right\}$. Since $U_{m}\left(a_{m}^{\prime },a_{-m}\right)<U_{m}\left(a_{m},a_{-m}\right)$ and $a_{m}=0,a_{m}^{\prime }>0$, it means that the device *m* changes to edge computing for reducing time overhead. According to Lemma 1, the interference of device *m* must satisfy a threshold $H_{m}$ that is $I_{m}^{a_{m}^{\prime }}<H_{m}$. Thus, case (2) is proved. For case (3), it is very similar to case (2). We use the same method in case (2) to do the subtraction:
(24)$$\begin{array}{c} \Phi_{m}\left(a_{m}^{\prime },a_{-m}\right)-\Phi_{m}\left(a_{m},a_{-m}\right)=\sum_{i=1,a=\left\{a_{m}^{\prime },a_{-m}\right\}}^{M}P_{i}h_{i}I_{i}^{a_{i}}+2P_{m}h_{m}H_{m}-\sum_{i=1,a=\left\{a_{m},a_{-m}\right\}}^{M}P_{i}h_{i}I_{i}^{a_{i}}\\ =2P_{m}h_{m}H_{m}-P_{m}h_{m}I_{m}^{a_{m}}+\sum_{i=1,i\neq m}^{M}P_{i}h_{i}\left(\sum_{j=1,j\neq i,j\neq m}^{M}\zeta_{j}^{a_{i}}P_{j}h_{j}\right)-\sum_{i=1,i\neq m}^{M}P_{i}h_{i}\left(\sum_{j=1,j\neq i,j\neq m}^{M}\zeta_{j}^{a_{i}}P_{j}h_{j}+\zeta_{i}^{a_{m}}P_{m}h_{m}\right)\\ =P_{m}h_{m}I_{m}^{a_{m}^{\prime }}-2P_{m}h_{m}H_{m}+\sum_{i=1,i\neq m}^{M}P_{i}h_{i}\left(\sum_{j=1,j\neq i,j\neq m}^{M}\zeta_{j}^{a_{i}}P_{j}h_{j}+\zeta_{i}^{a_{m}^{\prime }}P_{m}h_{m}\right)-\sum_{i=1,i\neq m}^{M}P_{i}h_{i}\left(\sum_{j=1,j\neq i,j\neq m}^{M}\zeta_{j}^{a_{i}}P_{j}h_{j}\right)\\ =2P_{m}h_{m}\left(H_{m}-I_{m}^{a_{m}}\right) \end{array}$$Similar to case (2), when the device change to local computing, we have $I_{m}^{a_{m}}>H_{m}$. Then, case (3) is proved. Combining results in the three cases, we can conclude that the edge computation offloading game is an ordinary potential game with potential function as given in (Equation 19). Then, Theorem 1 is proved. □

In this section, the definition of the ECOG is given. In addition, the properties of ECOG have been determined. Then, we will use the properties of the potential game to find the solution by Q-learning.

## 5. Distributed Q-learning Algorithm for Computation Offloading

As a popular algorithm in a channel allocation problem, Q-learning is often used in combination with game theory. In the smart city, each device’s perception of the environment is very limited, and it is difficult to represent the state of the system. Therefore, we apply a distributed and stateless Q-learning to solve the computation offloading problem. In this part, we will introduce a distributed reinforcement learning algorithm called stateless Q-learning and give two learning policies.

### 5.1. Stateless Q-Learning

In the smart city, it is so difficult for IoT devices to observe and record the state (such as other devices’ addictions) for the system. They can only sample and estimate their environment. Due to the small amount of information available to the device, a simple stateless variation of Q-learning, as formulated in [27], is used for our problem. In a smart city system with a lot of IoT devices, this reinforce learning algorithm is distributed and all IoT devices do not need to exchange information with others, resulting high efficiency.

In the Q-learning procedure, we formulate the Q-value for device *m* at time *k* as below:(25)$$\forall i\in A,\;Q_{m}^{i}\left(k\right)=\left\{\begin{array}{cc} \left(1-\alpha_{m}\left(k\right)\right)Q_{m}^{i}\left(k-1\right)+\alpha_{m}\left(k\right)r_{m}^{i}\left(k\right) & i=a_{m}\left(k\right)\\ Q_{m}^{i}\left(k-1\right) & otherwise \end{array}\right.$$
where $\alpha_{m}\left(k\right)$ is the learning rate of device *m* at time *k*, and $a_{m}\left(k\right)$ is the action taken by device *m* at time *k*. $r_{m}^{i}\left(k\right)$ is the reward that the device *m* chooses to take action *i* in time *k*. Based on Section 3, we define $r_{m}^{i}\left(k\right)$ as:(26)$$r_{m}^{i}\left(k\right)=TH_{m}-T_{m}\left(k\right)$$

$T_{m}\left(k\right)$ is the expected time for device *m* to finish task at time *k*. It means that, if the task is overtime, the IoT device will get a negative reward. Otherwise, the less time consumed, the greater reward for the devices. For each IoT device, they run the same distributed learning algorithm without synchronization. When distributed Q-learning start, each IoT device selects an action following to a policy in each iteration, then gets a reward to update the Q-table, and finally converges to a solution. Next, two Q-learning algorithms of different learning policies will be introduced, and the analysis of convergence (to NE) will be given.

### 5.2. Distributed Q-Learning with ϵ-Greedy Learning Policy

The ϵ-greedy learning policy is common in Q-learning. In our ECOG, the actions of devices in the Q-learning will converge to NE by taking ϵ-greedy policy. To show this policy, we need to define the $\epsilon_{k}$ as:(27)$$\epsilon_{k}=\epsilon_{0}k^{-1/M},\epsilon_{0}\in \left(0,1\right)$$

In ϵ-greedy selection policy, an IoT device *m* selects an action with maximum Q-value at time *k* with probability $\left(1-\epsilon_{k}\right)$ and chooses other actions randomly with probability $\epsilon_{k}$. It can be expressed as:(28)$$a_{m}\left(k\right)=\left\{\begin{array}{ll} \arg\max_{i}\left\{Q_{m}^{i}\left(k-1\right)\right\} & w.p.(1-\epsilon_{k})\\ \mathrm{randomly}\;\mathrm{selected}\;\mathrm{from}[0,N] & w.p.\epsilon_{k} \end{array}\right.$$

In addition, in order for the algorithm to converge to NE, the learning rate must follow the condition that:(29)$$\sum_{k=1}^{\infty }\alpha_{m}\left(k\right)=\infty \;and\;\sum_{k=1}^{\infty }{\left(\alpha_{m}\left(k\right)\right)}^{2}<\infty$$

Thus, we can achieve that by define the $\alpha_{m}\left(k\right)$ as:(30)$$\alpha_{m}\left(k\right)={\left[\beta +\Delta_{m}^{k}\left(a_{m}\left(k\right)\right)\right]}^{-\rho }$$
where $\Delta_{m}\left(a_{m}\left(k\right)\right)$ is the number of times the action $a_{m}\left(k\right)$ has been selected by device *m* up to time *k*. ρ∈(0.5,1] is the rate parameter. When ρ increases, the learning rate decreases faster, so the algorithm converges faster. In addition, β is a positive constant associated with the initial value of the learning rate. When β increases, the learning rate increases and the algorithm will converge faster. Note that the other learning rate that follows the condition (Equation 29) is feasible for the distributed stateless Q-learning. We just take a special case based on our experiences. In addition, since it is difficult to judge whether the algorithm converges, we define a sufficiently large number of iterations $k^{*}$. The algorithm will stop after $k^{*}$ iterations. After defining the relevant formulas, the summarization of the distributed Q-learning with ϵ-greedy learning policy is shown in Algorithm 1. Because the algorithm is distributed, there is no need to be synchronized. This reduces the signaling transmission in the iterative process and greatly improves the efficiency of the algorithm. For the result, if each IoT device in the smart city takes the Q-learning algorithm with an ϵ-greedy learning policy, the actions will converge to an NE point of an ECOG. Now, we will prove it.
**Algorithm 1** Distributed Q-learning for device *m* with ϵ-greedy learning policy.**for**i=0 to *N*
**do**    $Q_{m}^{i}\left(0\right)=0$**end for****for**k=1 to $k^{*}$
**do**    $\epsilon_{k}=\epsilon_{0}k^{-1/N}$    Follow the ϵ-greedy policy (Equation 28) to get the next action $a_{m}\left(k\right)$.    Based on $a_{m}\left(k\right)$, change the computing state and sub-channel.    Observe sub-channel interference and calculate the reward $r_{m}^{a_{m}\left(k\right)}\left(k\right)$ according to (Equation 26).    **for**
i=0 to *N*
**do**         Calculate the $Q_{m}^{i}\left(k\right)$ according to (Equation 25) with the learning rate formulated at (Equation 30)    **end for****end for**

**Theorem** **2.**
*For an ECOG, consider a distributed stateless Q-learning with an ϵ-greedy learning policy. Each time, the device selects the action according to (Equation 28) and updates the q-table according to (Equation 25) and (Equation 30). Then, for a sufficiently large number of iterations, k, the actions of all devices will converge to NE with probability 1.*


**Proof.** At the beginning, we define that a strategic form game is a weakly acyclic game (WAG) if, for any joint strategy, there exists a finite improvement path that starts at it. It is clearly that every OPG is weakly acyclic game [28]. Thus, ECOG is also a kind of WAG. Then, the result can be proved in three parts. First, we need to show that the process is weakly ergodic. Second, prove that the process is strongly ergodic. Finally, using the nature of WAG, it is proved that actions can only converge to NE. The proof is so complicated that it is omitted here. See [29] for details of the proof. □

Theorem 2 shows the stability of Q-learning. Regardless of the environment, stateless Q-learning with ϵ-greedy learning policy will certainly make ECOG converge to NE and give a relatively good solution. This is of great help to the problem of computing unloading in the smart city.

### 5.3. Distributed Q-Learning with Boltzmann Learning Policy

Compared with ϵ-greedy learning policy, Boltzmann learning policy is more sensitive to Q-values. Devices have a higher probability of choosing actions that have a higher Q-value. This policy, which has an obvious preference for the selection of Q-values, makes the distributed Q-learning with this learning policy have excellent convergence speed. In Boltzmann learning policy, each device needs to maintain an action selection vector $p_{m}\left(k\right)$ as shown below:(31)$$p_{m}^{i}\left(k\right)=\frac{e^{Q_{m}^{i}\left(k\right)/\rho_{m}\left(k\right)}}{\sum_{j=0}^{N}e^{Q_{m}^{j}\left(k\right)/\rho_{m}\left(k\right)}},\;\forall i\in A$$
where $\rho_{m}\left(k\right)$ is called virtual temperature depended on the cooling function [30]. In our algorithm, it is expressed as:(32)$$\rho_{m}\left(k\right)=\frac{\rho_{0}}{{\left(\log_{2}\left(k+1\right)\right)}^{\min\left\{2.5,(2M+k-1)/(2M)\right\}}}$$

This cooling function is based on experience and simulation, where $\rho_{0}$ is a positive constant called initial temperature. The temperature decreases slowly at first and then rapidly, resulting in the convergence of the actions. Specifically, in the process of algorithm execution, IoT devices need to choose actions according to its action selection vector which is determined by (Equation 31). At the beginning, the virtual temperature is high so that the selection probability of each action is close, which leads to more explorations of actions. This means more accurate estimates of the environment. Then, as the temperature gradually decreases, the device will be inclined to choose an action with a higher Q-value. In addition, the learning rate for this algorithm is a positive constant less than 1, that is,
(33)$$\alpha_{m}\left(k\right)=A0,\;0<A0<1$$

Based on the above description, pseudo code of distributed Q-learning with Boltzmann learning policy is shown as Algorithm 2.
**Algorithm 2** Distributed Q-learning for device *m* with Boltzmann learning policy.**for**i=0 to *N*
**do**    $Q_{m}^{i}\left(0\right)=0$    $p_{m}^{i}\left(0\right)=1/\left(N+1\right)$**end for****for**k=1 to $k^{*}$
**do**     **if**
$\max p_{m}^{i}\left(k\right)\geq 0.99$
**then**       **exit**.    **end if**    Choose the action $a_{m}\left(k\right)$ according to its current action selection probability vector $p_{m}\left(k\right)$.    Based on $a_{m}\left(k\right)$, change the computing state and sub-channel.    Observe sub-channel interference and calculate the reward $r_{m}^{a_{m}\left(k\right)}\left(k\right)$ according to (Equation 26).    **for**
i=0 to *N*
**do**         Calculate the $Q_{m}^{i}\left(k\right)$ according to (Equation 25) with the learning rate formulated at (Equation 33)         Update the action selection probability vector according to (Equation 31).    **end for****end for**

Since the virtual temperature will decrease, the probability of choosing the action with the largest Q-value will increase and the action of the device must be convergent. In order to prevent a dead cycle, the algorithm will stop when the *k* reaches the maximum number of iterations $k^{*}$. In our simulation, Q-learning with Boltzmann learning policy generally converges very quickly with a good result.

## 6. Simulation Result

In this section, some numerical simulation results will be provided to evaluate the performance and convergence of the distributed Q-learning algorithm. In our simulation, the MEC server will be located in the central of an area. There are *M* IoT devices randomly located around the MEC server. Without loss of generality, it is assumed that n=5 wireless sub-channels are available in this area. The sub-channel bandwidth is 0.8, 1, 2, 5, 10 Mhz, respectively. For the IoT devices, we assume that different devices have different computation capabilities. The parameters of IoT devices and MEC server are shown in Table 1.

To get closer to a real smart city, we designed five different computing tasks. Each task has its own characteristics, as shown in Table 2.

Since there are so many applications and services in the smart city, it is considered that each task is a real task in the smart city. For example, task 3 has a large size input and requires a lot of computation, which is very similar to the human activity recognition task. Task 4 only requires a small amount of calculation to complete, and we can think of it as a simple task of uploading and synchronizing traffic information.

In addition, for a Q-learning algorithm with ϵ-greedy learning policy, $\epsilon_{k}$ is equal to 0.125 and its learning rate is $\alpha_{m}\left(k\right)={\left[0.5+\Delta_{m}^{k}\left(a_{m}\left(k\right)\right)\right]}^{-0.6}$ for all m∈M. In addition, for Q-learning with Boltzmann learning policy, it has an initial virtual temperature $\rho_{0}=3$ and a fixed learning rate $\alpha_{m}\left(k\right)=0.3$. To evaluate the performance of the two algorithms, we introduce another distributed computation offloading algorithm (DCOA) [14] for comparison. The DCOA is based on the FIP of the potential game. In each iteration, the MEC server will send the decisions of all devices to each device, and each device will calculate the best decision it can get if the other devices do not change their decisions. MEC server will then randomly select only one device to change its decision. When all the devices can not find the next best decision, the DCOA stops. We also introduce all local computing for comparison, that is, all devices only complete tasks locally. Next, we will show our simulation results and give the analysis.

Firstly, to evaluate the performance of the result, for a certain *M*, we randomly generated 1000 sets of test parameters, and run the four methods using these parameters. All algorithms are limited to 200 iterations. The results of *M* are 5, 10, 20, 30, 40, and 50 are shown in Figure 2 and Figure 3. As shown in Figure 2, Q-learning algorithms perform well in terms of average time overhead. When M=5, the average time overhead of strategies obtained by DCOA is the lowest. The results of Q-learning are not as good as DCOA, suggesting that DCOA has an advantage in scenarios with a small number of IoT devices. However, when M>5, Q-learning can get the strategy of less time consumption. Moreover, with the increase of the number of access devices, by using the Q-learning algorithm, the delay does not increase significantly, and the system can still run smoothly without causing congestion. Overall, with MEC, the average time consumption of devices using Q-learning algorithm is 23% to 51% less than that of local computing. Then, we define that on-time rate is the ratio of the number of devices that meet the threshold. In addition, in Figure 3, the distributed Q-learning algorithm allows more devices to meet the threshold. When the number of devices is small, the Q-learning algorithm using Boltzmann learning policy has a good performance. However, with a large number of devices, Q-learning with ϵ-greedy learning policy works better. The on-time rate obtained by Q-learning is about 0.1 higher than that of DCOA, which indicates the good performance of Q-learning.

Second, we will evaluate the convergence process of the algorithms. In this part, for M=10 and M=30, each algorithm will run 1000 times in one set of parameters. We forced all the algorithms to iterate 200 times. Then, take the average of the results after each iteration, and Figure 4 and Figure 5 are obtained. These figures are the mean values of the results after many experiments, which can reflect the general convergence trend of the algorithms. However, they can not represent the exact speed of convergence. According to the figures, whether M=10 or M=30, both Q-learning algorithms rapidly converge to a solution. The process of convergence is the same as the description in Section 5. In Figure 4 and Figure 5, Q-learning algorithms show a good performance in on-time rate and average time overhead. When M=10, Q-learning using Boltzmann policy can achieve less time overhead and higher on-time rate. When M=30, the performance of the two Q-learning is almost the same. In addition, in both scenarios, Q-learning performs better than DCOA. The solution obtained by Q-learning algorithm converges to NE, which is a stable state acceptable to all devices. In addition, the NE point obtained by Q-learning is better than that obtained by DCOA. Note that the time overhead and on-time rate of Q-learning fluctuated rather than changed monotonously. As Q-learning algorithm is based on learning, it will explore different solutions in the iterative process. This causes the curve to fluctuate. Finally, both Q-learning algorithms converged and stoped the fluctuation.

In particular, compared to the Q-learning with ϵ-greedy policy, Q-learning with Boltzmann learning policy will stop immediately after reaching convergence, reducing meaningless iterations and improving algorithm efficiency. We can obtain the average iteration time required for convergence in the Q-learning with policy Boltzmann policy and DCOA. The results are shown as in Figure 6. In Figure 6, the average convergence times of Q-learning are related to the random exploration in each experiment, which can be used as a reference for the convergence speed in the actual smart city. It shows that the average number of times for convergence increases almost linearly as the number of IoT devices increases. In addition, the smaller $\rho_{0}$ is, the faster Q-learning converges. However, the convergence speed of Q-learning is significantly higher than that of DCOA because Q-learning needs exploration and learning to get a better result. However, DCOA needs to broadcast the decision of each IoT device, which requires more signaling exchange than Q-learning.

## 7. Conclusions and Future Work

In the smart city, thousands of IoT devices simultaneously generate massive computing tasks to keep multiple services running in the city. Virtual reality, high-definition video, live event broadcasting, and other IoT applications put forward higher requirements on task processing time. Facing this challenge, this paper has focused on finding the proper computation offloading strategies with MEC server adopted by the IoT devices in the smart city. We apply a distributed reinforcement learning algorithm called stateless Q-learning and give two learning policies. This algorithm has many advantages in the smart city scenario. Firstly, it has been found that the distributed Q-learning algorithm has good performance in reducing time overhead and convergence. Compared with local computation, with the MEC server, the Q-learning algorithm can reduce the average time overhead by up to 51%. In addition, compared with DCOA, the on-time rate obtained by Q-learning is about 0.1 higher. In addition, according to the mathematical proof and simulation results, Q-learning can converge to NE, which means that the algorithm can produce a solution that is acceptable to all IoT devices. This avoids the congestion caused by uneven distribution of computing resources. Moreover, the distributed algorithm only needs a small amount of signaling exchange. Therefore, it is suitable for the smart city with lots of IoT devices.

In our future work, we may consider more general scenarios in the smart city. For example, combining with the queuing theory, we can create a computing model more in line with the actual situation of the smart city. In addition, the upload, download, and control protocol transport time can be considered to make the communication model better. In addition, collaboration between multiple MEC servers is also worth considering. For the Q-learning algorithm, if the environment changes, it needs to iterate many times to get the NE point, resulting in a low efficiency. In the future, combined with the neural network or any other learning methods, it may improve the performance of the algorithm. 

## Figures and Tables

**Figure 1 sensors-20-00973-f001:**
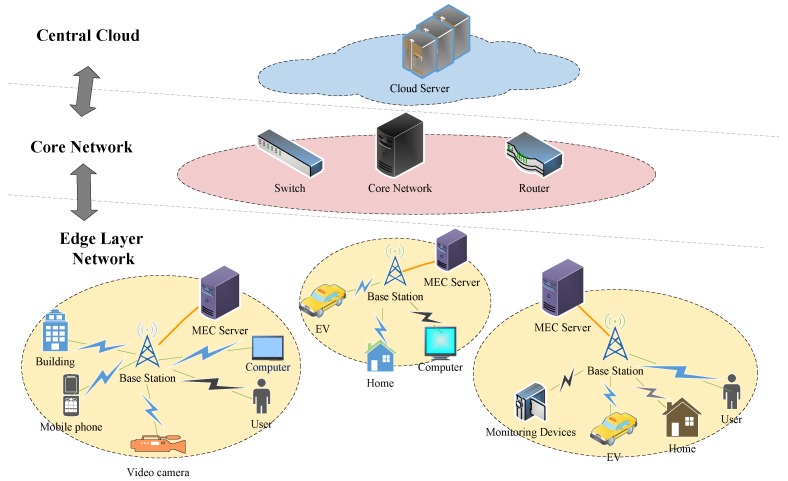
Applications of mobile edge computing in the smart city.

**Figure 2 sensors-20-00973-f002:**
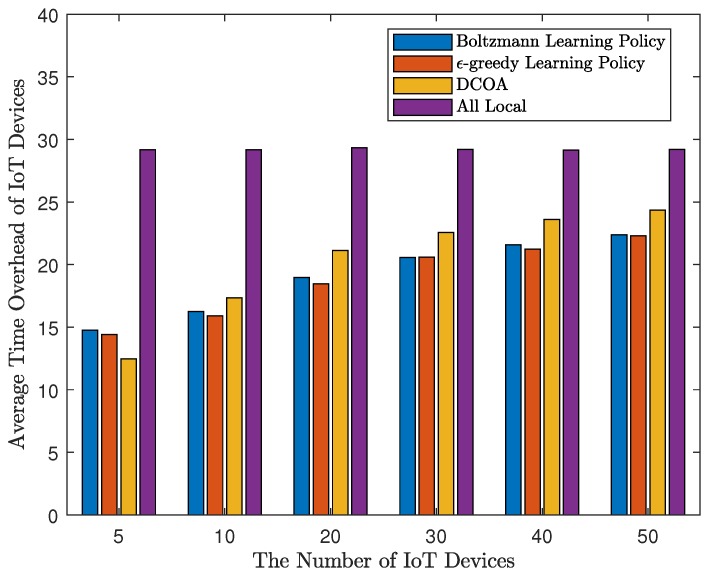
Average time overhead of the Internet of Things (IoT) devices.

**Figure 3 sensors-20-00973-f003:**
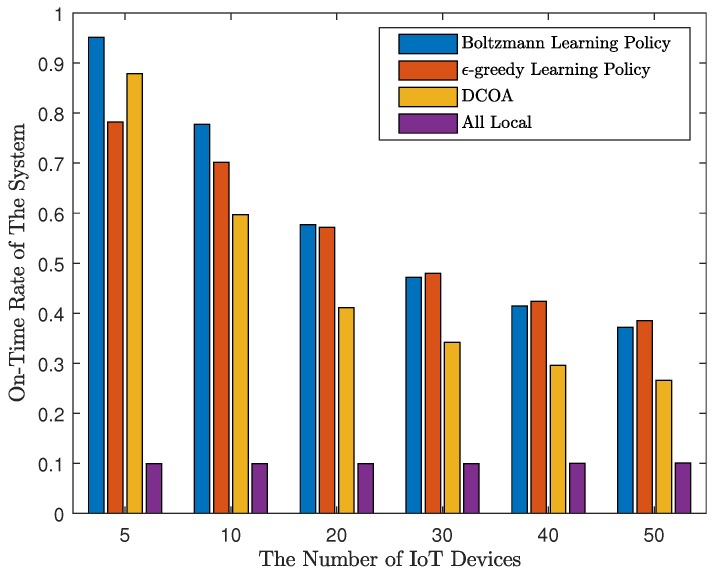
On-time rate of IoT devices.

**Figure 4 sensors-20-00973-f004:**
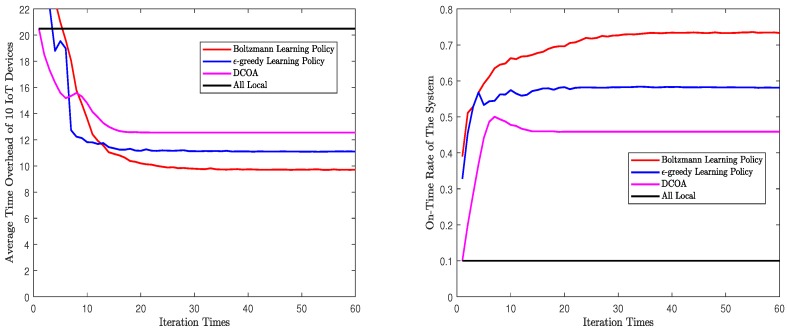
Convergence processes of different algorithms with 10 IoT devices.

**Figure 5 sensors-20-00973-f005:**
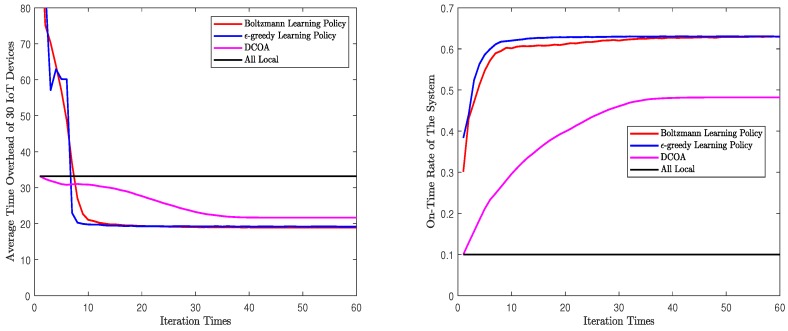
Convergence processes of different algorithms with 30 IoT devices.

**Figure 6 sensors-20-00973-f006:**
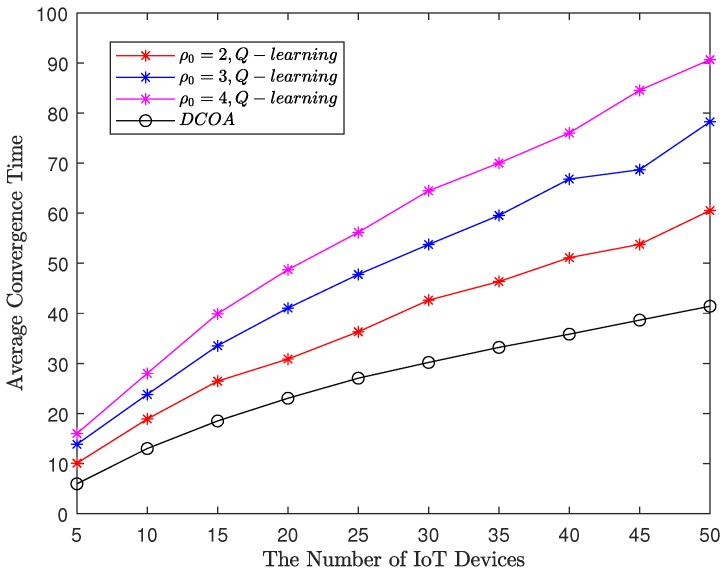
Convergence time of the Q-learning with Boltzmann learning policy and distributed computation offloading algorithm (DCOA).

**Table 1 sensors-20-00973-t001:** The parameters of Internet of Things (IoT) devices and mobile edge computing (MEC) server.

Parameter	Description	Value	Unit
PL	Pathloss	35.3+37.6∗log10(d)	dB
$\delta^{2}$	Thermal noise	−179.0	dBm/Hz
*d*	Distance between devices and server	[200,800]	meter
$f_{i}$	Computation capability of devices	$\left\{2000,3300\right\}$	cycle/s
$f^{e}$	Computation capability of MEC server	7000	cycle/s
$P_{i}$	Transmit power of IoT devices	20	dBm

**Table 2 sensors-20-00973-t002:** The tasks in the smart city

Task Type *i*	$D_{i}$	$C_{i}$	${\mathrm{TH}}_{i}$
0	7000 Kb	20,000 cycles	7 s
1	4000 Kb	82,000 cycles	22 s
2	2200 Kb	150,000 cycles	45 s
3	10,000 Kb	100,000 cycles	28 s
4	5000 Kb	10,000 cycles	3 s

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
