# Peer review of "Distributed Learning Based Joint Communication and Computation Strategy of IoT Devices in Smart Cities"

_sensors, 2020, doi:10.3390/s20040973_

Round 1
Reviewer 1 Report
The goal of this study was to find the proper computation offloading strategies adopted by the IoT devices in the smart city. Mainly, the uplink of wireless network integrated with multiple IoT devices was considered and it is assumed that IoT devices are associated with one MEC server. This study focuses on finding a proper computing strategy, including computing state and wireless sub-channel selection, to minimize the expected time to accomplish tasks meeting the threshold.
Furthermore, based on the system model, a computation offloading game with IoT devices was formulated, and it was proved that the game has at least one Nash equilibrium point. I was proposed two distributed Q-learning algorithms with different learning policies to reach the Nash equilibrium.
In addition, it has been found that the distributed Q-learning algorithm has good performance in reducing time overhead and convergence. This shows that the algorithms have good convergence speed.
This article has a good structure and a well-explained methodology. All the technical terms are well-explained so that any reader could understand the concepts. This paper specifically discusses about the joint policy of communication and computing of IoT devices in edge computing through game theory, and proposes distributed Q-learning algorithms with two learning policy. Simulation results show that the algorithm can converge quickly with a balanced solution.
The article needs to be reviewed for spelling and grammar errors:
- avarage time - average time
- it need - it needs
- to iterate for many times - to iterate many times
- perfromance - performance
The references should include related work about time critical IoT applications:
- Štefanič, Polona, Matej Cigale, Andrew C. Jones, Louise Knight, Ian Taylor, Cristiana Istrate, et al. "SWITCH workbench: A novel approach for the development and deployment of time-critical microservice-based cloud-native applications." Future Generation Computer Systems 99 (2019): 197-212.
- Kakderi, Christina, Nicos Komninos, and Panagiotis Tsarchopoulos. "Smart cities and cloud computing: lessons from the STORM CLOUDS experiment." Journal of Smart Cities 1, no. 2 (2019): 4-13.
- Ciobanu, Radu-Ioan, Ciprian Dobre, Mihaela Bălănescu, and George Suciu. "Data and Task Offloading in Collaborative Mobile Fog-Based Networks." IEEE Access 7 (2019): 104405-104422.
- Sapienza, Marco, Ermanno Guardo, Marco Cavallo, Giuseppe La Torre, Guerrino Leombruno, and Orazio Tomarchio. "Solving critical events through mobile edge computing: An approach for smart cities." In 2016 IEEE International Conference on Smart Computing (SMARTCOMP), pp. 1-5. IEEE, 2016.
Reviewer 2 Report
The paper addresses a Distributed Learning Based Joint Communication and
Computation Strategy in the IoT environment of a Smart City. The authors propose a structured approach to addressing IoT in Smart City environment but although the presentation is convincing the impact on the research is unclear. The authors should better address the state of the art section and clearly point out the advantages of the proposed system. Another key problem of the paper is related to the fact that the focus is general and not correlated with the real characteristics of a Smart City.
The following issues must be addressed before publishing:
The state of the art research in the domain should be extended. The context of Smart City and the relation to IoT are insufficiently described. The authors need to better define the concept of Smart City. Also some relevant case studies and be analysed. The impact of the research is not clearly presented in relation to Smart City. The characteristics of a Smart City need to be clearly addressed and measured. The experimental method is not clearly described from the perspective of the Smart City. The experimental results are not clearly analysed in comparison with other results from other researchers. It is not clear what are the advantages of such an approach. The conclusions should point out the advantages of using such an approach in correlation with the smart city.
Reviewer 3 Report
The authors have made an interesting attempt to write a paper addressing a distributed learning based joint communication and computation strategy of IoT devices in smart city. Introduction: From my point of view, introduction is not well focused. In a research paper, it is expected that introduction section briefly explains the starting background and, even more important, the originality (novelty) and relevancy of the study is well established. Once this is done, hypothesis and objectives of the study need to be addressed, as well as a brief justification of the conducted methodology. It is my belief that, in this case, authors do not put effort enough (or any effort) in highlighting the relevancy and (specially) the novelty of the study. Consequently, both major aspects are compromised. I strongly recommend that authors clearly explain all these aspects (including hypothesis and objectives) in order to add scientific rigor to the manuscript. Discussion: It is my opinion that a separate discussion section would help the reader to understand the study. However, the main issue arises from the lack of comparison with state-of-the-art studies. From my point of view, even though some publications are listed in the literature review, it has no justification not conducting a comparison with main previous works already published. The lack of such a comparison compromises the significance of the paper, so I strongly recommend authors to conduct as much and suitable comparisons as needed to solve this issue. On page 10, the authors claim that “β is a positive constant” – line 272 – what constant is β? Please explain in detail. Discussion and Result Analysis should be addressed in more detail. For instance, in Figure 6, ρ0 behaves differently form 40 to 45 IoT devices. The conclusion is incomplete. The obtained results should be briefly and clearly mentioned through the support of numerical data. What were the most sounding quantifiable findings of this study? The conclusion is intended to help the reader understand why your research should matter to them after they have finished reading the paper. A conclusion is not merely a summary of the main topics covered or a re-statement of your research problem, but a synthesis of key points and, if applicable, where you recommend new areas for future research. This paper seems to be fairly crafted, yet with some limitations. Thus, the authors need to improve in order to have an acceptable paper for Sensors.Author Response
Please see the attachment.

Reviewer 4 Report
The approach is well-described and the results are interesting.
The related works section should be added. It should provide information about the differences between the cited solutions and the proposed paper, exposing the improvements introduced. Un this direction, authors could consider such recent works: Event-based sensor data exchange and fusion in the Internet of Things environments., in Journal of Parallel and Distributed Computing, 2018; Internet of things reference architectures, security and interoperability: A survey, Internet of Things, 2018.
Finally, typos and grammar have to be revised.
The organization of contents is good and their description is quite fluent.
